# Antimicrobial resistance and AmpC production in ESBL-producing *Klebsiella pneumoniae* and *Klebsiella quasipneumoniae*: A retrospective study in Japanese clinical isolates

**Naoki Watanabe**[1,2]*, **Tomohisa Watari**[1], **Yoshihito Otsuka**[1], **Masahiko Ito**[3], **Kazufumi Yamagata**[2], **Miyuki Fujioka**[2]

**1** Department of Clinical Laboratory, Kameda Medical Center, Kamogawa, Chiba, Japan, **2** Graduate School of Health Sciences, Hirosaki University, Hirosaki, Aomori, Japan, **3** Sapporo Clinical Laboratory Inc., Chuo-ku, Sapporo, Hokkaido, Japan

* watanabe.naoki.4@kameda.jp

## Abstract

### Introduction

The study of *Klebsiella quasipneumoniae*, *Klebsiella variicola*, and AmpC production in extended-spectrum β-lactamase (ESBL)-producing *Klebsiella* in Japan is limited, and existing data are insufficient. This study aims to characterize *Klebsiella* species, determine AmpC production rates, and analyze antimicrobial resistance patterns in ESBL-producing *Klebsiella* isolates in Japan.

### Methods

A total of 139 clinical isolates of ESBL-producing *Klebsiella* were collected in Japan, along with their corresponding antimicrobial susceptibility profiles. The isolates were identified using a web-based tool. ESBL genes within the isolates were identified using multiplex PCR. Screening for AmpC-producing isolates was performed using cefoxitin disks, followed by multiplex PCR to detect the presence of AmpC genes. Antimicrobial resistance patterns were analyzed across the predominant ESBL genotypes.

### Results

The web-based tool identified 135 isolates (97.1%) as *Klebsiella pneumoniae* and 4 (2.9%) as *K. quasipneumoniae* subsp. *similipneumoniae*, with no instances of *K. variicola* detected. Among *K. pneumoniae*, the CTX-M-1 group emerged as the predominant genotype (83/135, 61.5%), followed by *K. quasipneumoniae* subsp. *similipneumoniae* (3/4, 75.0%). The CTX-M-9 group was the second most prevalent genotype in *K. pneumoniae* (45/135, 33.3%). The high resistance rates were observed for quinolones (ranging from 46.7% to 63.0%) and trimethoprim/sulfamethoxazole (78.5%). The CTX-M-1 group exhibited higher

**Data Availability Statement:** All relevant data are within the manuscript and its Supporting information files.

**Funding:** The author(s) received no specific funding for this work.

**Competing interests:** The authors have declared that no competing interests exist.

resistance to ciprofloxacin (66/83, 79.5%) compared to the CTX-M-9 group (18/45, 40.0%), a trend also observed for levofloxacin and trimethoprim/sulfamethoxazole. Among the 16 isolates that tested positive during AmpC screening, only one *K. pneumoniae* isolates (0.7%) were confirmed to carry the AmpC gene.

## Conclusion

*Klebsiella pneumoniae* with the CTX-M-1 group is the most common ESBL-producing *Klebsiella* in Japan and showed a low proportion of AmpC production. These isolates are resistant to quinolones and trimethoprim/sulfamethoxazole, highlighting the challenge of managing this pathogen. The findings underscore the importance of broader research and continuous monitoring to address the resistance patterns of ESBL-producing *Klebsiella*.

## Introduction

*Klebsiella pneumoniae* is a Gram-negative bacillus belonging to the Enterobacteriaceae family [1]. Recently, multidrug-resistant *K. pneumoniae* has emerged as a global concern. *Klebsiella pneumoniae* isolates that are resistant to extended-spectrum β-lactamases (ESBL) and carbapenems have been recognized as significant public health threats. Various clones, including clonal groups 258, 14 [2], ST307, and 147 [3,4] are associated with multidrug-resistant *K. pneumoniae*.

In ESBL-producing *K. pneumoniae*, the CTX-M-1 group, particularly CTX-M-15, is the most common ESBL genotype. CTX-M-15 is prevalent in various regions, including the United States [5], Canada [6], Europe [7,8], and Africa [9]. Among the isolates of ESBL-producing *K. pneumoniae* in Japan, CTX-M-15 is the predominant genotype [10]. Moreover, ESBL-producing isolates can co-produce AmpC. A study conducted in Iran reported a high prevalence of co-producing AmpC isolates in ESBL-producing *K. pneumoniae* [11]. However, research on ESBL and AmpC co-producing *K. pneumoniae* is limited in Japan.

*Klebsiella pneumoniae* has closely related species, such as *Klebsiella variicola* and *Klebsiella quasipneumoniae*, which include subspecies like *K. quasipneumoniae* subsp. *quasipneumoniae* and *K. quasipneumoniae* subsp. *similipneumoniae* [12]. ESBL production has also been observed in *K. quasipneumoniae* and *K. variicola* [13,14]. However, *K. quasipneumoniae* and *K. variicola* are frequently misclassified as *K. pneumoniae* in clinical settings [15]. Consequently, the epidemiology of ESBL-producing *K. quasipneumoniae* and *K. variicola* remains poorly understood in Japan.

In this study, we aimed to characterize *Klebsiella* species, ESBL genotypes, antimicrobial resistance patterns, and AmpC co-production in ESBL-producing *Klebsiella* isolates in the eastern and northern regions of Japan. Additionally, we compared antimicrobial resistance patterns among different *Klebsiella* species and major ESBL genotypes. Our findings provide updated insights into ESBL-producing *Klebsiella* and their antimicrobial resistance patterns. These insights will be invaluable for clinicians and researchers in understanding the dynamics of ESBL-producing *K. pneumoniae* and the characteristics of *K. quasipneumoniae* and *K. variicola*.

## Material and methods

We conducted a retrospective study on ESBL-producing *Klebsiella* in Japan. This study involved only isolates and their associated data without including any patient information linked to the isolates. Consequently, the Research Ethics Committee of the Kameda Medical

Center in Japan, determined that ethical review and informed consent were not required for this research.

## Isolate collection

In total, we collected 139 ESBL-producing *Klebsiella* isolates from two locations in Japan: Kameda Medical Center in the eastern part and Sapporo Clinical Laboratory in the northern part. At Kameda Medical Center, 90 isolates were obtained between February 2020 and November 2023, while 49 isolates were collected at Sapporo Clinical Laboratory from October to November 2023. All isolates were isolated during routine testing at each laboratory and stored at -80˚C until the study commenced. Only the initial isolate was incorporated into the study when multiple ESBL-producing *K. pneumoniae* isolates were derived from the same patient. Isolates collected before October 2023 were not preserved at the Sapporo Clinical Laboratory and were excluded from this study. These isolates were identified as either *K. pneumoniae* or *K. variicola* using matrix-assisted laser desorption ionization-time of flight mass spectrometry (Bruker Daltonik GmbH, Germany). The ESBL phenotype was confirmed using the Double-disk synergy test [16,17]. Before the *Klebsiella* MALDI TypeR [18], ESBL, and AmpC testing, isolates were cultured on Trypticase Soy Agar with 5% Sheep Blood (Becton Dickinson, USA) under aerobic conditions.

## Identification of *Klebsiella* isolates

Routine testing often does not distinguish between closely related *Klebsiella* species such as *K. quasipneumoniae* and *K. variicola*. To address this, we used the *Klebsiella* MALDI TypeR [18], a web-based tool for identifying *Klebsiella*. The tool acquires mass spectra of isolates and analyzes them through a web-based platform. A single colony was selected using a toothpick and applied to a target plate. Then, one μL of matrix solution (α-cyano-4-hydroxycinnamic acid [Bruker Daltonik GmbH], dissolved in acetonitrile/water/trifluoroacetic acid [50:47.5:2.5 v/v, Sigma-Aldrich, USA]), was added to the colony smear and allowed to air-dry. Mass spectra were acquired using a microflex LT/SH and flexControl Version 3.4 (Bruker Daltonik GmbH). Species identification was conducted by uploading the mass spectra to https://maldityper.pasteur.fr/ for analysis.

## Antimicrobial susceptibility testing

Antimicrobial susceptibility testing was conducted using the broth dilution method with an EIKEN dry plate (Eiken Chemical, Japan). In this assay, the bacterial solution was inoculated into microwells containing antibiotics. Following an incubation period, bacterial growth was assessed to ascertain the minimum inhibitory concentration (MIC) for each antibiotic. The test encompassed 14 antibiotics, including amoxicillin-clavulanate, piperacillin-tazobactam, ceftazidime, ceftriaxone, cefepime, cefmetazole, imipenem, meropenem, gentamicin, tobramycin, amikacin, ciprofloxacin, levofloxacin, and trimethoprim/sulfamethoxazole (TMP-SMX). The assay conditions and interpretation of results were adherent to the Clinical and Laboratory Standards Institute M100 [19]. Quality control was assured using the *Escherichia coli* ATCC 25922 isolate. Data on Antimicrobial susceptibility was collected and validated after confirming the absence of any discrepancies in the quality control process.

## Screening test for AmpC production

The screening test for AmpC was conducted using the cefoxitin disk susceptibility test as outlined by the European Committee on Antimicrobial Susceptibility Testing [20]. BD BBL Sensi-

Disc Cefoxitin FOX-30 (Becton Dickinson, USA) was utilized, and a zone diameter of less than 19 mm around the cefoxitin disc indicated a positive result for AmpC screening.

## Detection of ESBL and AmpC genotypes

Template DNA was prepared using the Cica GeneusR DNA Extraction Reagent (Kanto Chemical, Japan). The ESBL and AmpC genotypes were confirmed using the Cica GeneusR ESBL Genotype Detection Kit 2 and Cica GeneusR AmpC Genotype Detection Kit 2, respectively (both from Kanto Chemical). These kits are commercial products based on multiplex PCR for targeting ESBL or AmpC genes and were utilized following the manufacturer's instructions (Kanto Chemical). The Cica GeneusR ESBL Genotype Detection Kit 2 targets ESBL genes, such as the CTX-M group (M-1, M-2, M-8, M-9, M-25, and M chimera), SHV, GES, and TEM. Meanwhile, the Cica GeneusR AmpC Genotype Detection Kit 2 targets AmpC genes, including the CIT, FOX, ACT, DHA, ACC, and MOX families. Multiplex PCR was performed using the GeneAtlas TypeG thermal cycler (Astec, Japan). Primers provided with the kits were used, and the PCR protocol involved 30 cycles at 94 ˚C for 15 s, 63 ˚C for 15 s, and 72 ˚C for 40 s. PCR products were electrophoresed on a 2.0% agarose gel in $1 \times$ Tris-acetate-EDTA buffer and stained with 1.0% ethidium bromide.

## Statistical analysis

Statistical analysis was conducted using EZR version 1.54 [21], with statistical significance set at $P < 0.05$. We calculated the percentage and 95% confidence interval (95% CI) for the proportion of each *Klebsiella* species and AmpC-producing isolates within the ESBL-producing *Klebsiella*. The proportion of identified species within ESBL-*Klebsiella* was analyzed using the one-sample proportion test. Furthermore, comparisons of categorical variable ratios between two groups were performed using Fisher's exact test. This included analyzing the resistance ratio to each antibiotic for each *Klebsiella* species, the ratio of ESBL genotypes among each *Klebsiella* species, and the resistance ratio to each antibiotic among the major genotypes.

## Results

### Characterization of *K. pneumoniae*, *K. quasipneumoniae*, and *K. variicola* among the ESBL-producing *Klebsiella* isolates

The isolation specimens and characteristics of the ESBL-producing *Klebsiella* isolates are presented in Table 1. Most of the ESBL-*Klebsiella* isolates were obtained from urine specimens (74/139, 53.2%), followed by sputum (35/139, 25.2%), and blood (15/139, 10.8%). Using *Klebsiella* MALDI TypeR for identification, 135 isolates (97.1%, [95% CI, 92.8–99.2%]) were classified as *K. pneumoniae* and four (2.9%, [95% CI, 0.8–7.2%]) as *K. quasipneumoniae* subsp. *similipneumoniae*. There were significant differences in ESBL-*Klebsiella* between *K. pneumoniae* and *K. quasipneumoniae subsp. similipneumoniae* ($p < 0.001$). *K. variicola* was not detected in the ESBL-producing *Klebsiella* isolates (95% CI, 0.0–2.6%). All ESBL-*Klebsiella* isolates tested positive in the double-disk synergy test (139/139, 100%) and were found to harbor one to three β-lactamase genes. Detailed information on the isolates and *Klebsiella* MALDI TypeR results can be found in S1 Table.

### Antimicrobial susceptibilities of *K. pneumoniae* and *K. quasipneumoniae* subsp. *similipneumoniae*

The antimicrobial susceptibilities of *K. pneumoniae* and *K. quasipneumoniae* subsp. *similipneumoniae* are detailed in Table 2. *K. pneumoniae* had an antimicrobial resistance rate higher

**Table 1. Characteristics of ESBL-producing *K. pneumoniae* and *K. quasipneumoniae* subsp. *Similipneumoniae*.**

| Characteristics | Group of *Klebsiella* species | | | P value |
|---|---|---|---|---|
| | **All** (***n*** = 139) | ***K. pneumoniae*** (***n*** = 135) | ***K. quasipneumoniae*** (***n*** = 4) | |
| Specimen, no. (%) | | | | 0.7 |
| Urine | 74 (53.2) | 72 (53.3) | 2 (50.0) | |
| Sputum | 35 (25.2) | 35 (25.2) | 1 (25.0) | |
| Blood | 15 (10.8) | 14 (10.4) | 1 (25.0) | |
| Wound | 5 (3.6) | 5 (3.7) | 0 (0.0) | |
| Stool | 3 (2.2) | 3 (2.2) | 0 (0.0) | |
| Other* | 7 (5.0) | 7 (5.2) | 0 (0.0) | |
| Number of ESBL-gene, no. (%) | | | | < 0.001** |
| One | 7 (5.0) | 4 (3.0) | 3 (75.0) | < 0.001** |
| Two | 48 (34.5) | 47 (34.8) | 1 (25.0) | 1 |
| Three | 84 (60.4) | 84 (62.2) | 0 (0.0) | 0.02** |
| AmpC screening, positive no. (%) | 17 (12.2) | 16 (11.9) | 1 (25.0) | 0.4 |

*K. quasipneumoniae*, *K. quasipneumoniae* subsp. *similipneumoniae*; NA, not applicable.

* Other specimens included two ascites, two bile, two catheters, and one peritoneal dialysis fluid;

** P < 0.05.

**Table 2. Antimicrobial resistant isolates of ESBL-producing *K. pneumoniae* and *K. quasipneumoniae* subsp. *Similipneumoniae*.**

| Antibiotic | Resistant isolates, no. (%) | | P value |
|---|---|---|---|
| | ***K. pneumoniae*** (***n*** = 135) | ***K. quasipneumoniae*** (***n*** = 4) | |
| β-Lactam combination | | | |
| Amoxicillin-clavulanate | 7 (5.2) | 0 (0.0) | 1 |
| Piperacillin-tazobactam | 25 (18.5) | 0 (0.0) | 1 |
| Cephem | | | |
| Ceftazidime | 68 (50.4) | 3 (75.0) | 0.6 |
| Ceftriaxone | 131 (97.0) | 4 (100) | 1 |
| Cefepime | 96 (71.1) | 3 (75.0) | 1 |
| Cefmetazole | 4 (3.0) | 0 (0.0) | 1 |
| Carbapenem | | | |
| Imipenem | 0 (0.0) | 0 (0.0) | NA |
| Meropenem | 0 (0.0) | 0 (0.0) | NA |
| Aminoglycoside | | | |
| Gentamicin | 48 (35.6) | 0 (0.0) | 0.3 |
| Tobramycin | 23 (17.0) | 0 (0.0) | 1 |
| Amikacin | 0 (0.0) | 0 (0.0) | NA |
| Fluoroquinolone | | | |
| Levofloxacin | 63 (46.7) | 1 (25.0) | 0.6 |
| Ciprofloxacin | 85 (63.0) | 1 (25.0) | 0.2 |
| TMP-SMX | 106 (78.5) | 2 (50.0) | 0.2 |

*K. quasipneumoniae*, *K. quasipneumoniae* subsp. *similipneumoniae*; NA, not applicable; TMP-SMX, trimethoprim/sulfamethoxazole.

overall than *K. quasipneumoniae* subsp. *similipneumoniae*. The resistance rates for levofloxacin were 46.7% in *K. pneumoniae* (63/135) and 25% in *K. quasipneumoniae* subsp. *similipneumoniae* (1/4), while for ciprofloxacin, they were 63.0% in *K. pneumoniae* (85/135) and 25% in *K. quasipneumoniae* subsp. *similipneumoniae* (1/4). The resistance rates to TMP-SMX were 78.5% (106/135) in *K. pneumoniae* and 50.0% (2/4) in *K. quasipneumoniae* subsp. *similipneumoniae*. No significant differences in antimicrobial susceptibilities were noted between *K. pneumoniae* and *K. quasipneumoniae* subsp. *similipneumoniae*. Resistance to cefmetazole was found in 3% of *K. pneumoniae* isolates (4/135) but was absent in *K. quasipneumoniae* subsp. *similipneumoniae* (0/4).

## ESBL genotypes of *K. pneumoniae* and *K. quasipneumoniae* subsp. *similipneumoniae*

The ESBL genotypes of *K. pneumoniae* and *K. quasipneumoniae* are detailed in Table 3. ESBL genotypes detected in these isolates included CTX-M-1, CTX-M-2, and CTX-M-9, SHV, and TEM. The CTX-M-1 group was the most prevalent in both *K. pneumoniae* (83/135, 61.5%) and *K. quasipneumoniae* (3/4, 75.0%). Among the CTX-M-1 groups, CTX-M-1 alone was only observed in *K. quasipneumoniae* (3/4, 75.0%) and not in *K. pneumoniae* (P < 0.0001). The isolate harboring the CTX-M-1, SHV, and TEM was detected in 56.3% (76/135) of *K. pneumoniae* but was not found in *K. quasipneumoniae* (P = 0.04).

The CTX-M-9 group was exclusively present in *K. pneumoniae* (45/135, 33.3%) and not in *K. quasipneumoniae* (P = 0.3). Among the *K. pneumoniae* isolates, 27.4% (37/135) harbored the CTX-M-9 group and SHV, while 5.9% (8/135) harbored the CTX-M-9 group, SHV, and TEM. Comprehensive data on the isolates and patients are provided in S2 Table.

## Antimicrobial susceptibilities of CTX-M-1 and CTX-M-9 groups in *K. pneumoniae*

The antimicrobial susceptibilities of the CTX-M-1 and CTX-M-9 groups are summarized in Table 4. Amoxicillin-clavulanate, cefmetazole, and carbapenems displayed low resistance

**Table 3. ESBL genotypes of *K. pneumoniae* and *K. quasipneumoniae* subsp. *Similipneumoniae*.**

| ESBL genotype, % (no.) | Percentage of cohort (no.) | | P value |
|---|---|---|---|
| | *K. pneumoniae* (*n* = 135) | *K. quasipneumoniae* (*n* = 4) | |
| CTX-M-1 group only | 0 (0.0) | 3 (75.0) | < 0.001* |
| CTX-M-1 group + SHV | 7 (5.2) | 0 (0.0) | 1 |
| CTX-M-1 group + SHV + TEM | 76 (56.3) | 0 (0.0) | 0.04* |
| CTX-M-9 group + SHV | 37 (27.4) | 0 (0.0) | 0.6 |
| CTX-M-9 group + SHV + TEM | 8 (5.9) | 0 (0.0) | 1 |
| CTX-M-2 group + SHV | 3 (2.2) | 0 (0.0) | 1 |
| SHV only | 4 (3.0) | 0 (0.0) | 1 |
| SHV + TEM | 0 (0.0) | 1 (25.0) | 0.03* |
| CTX-M-8 group | 0 (0.0) | 0 (0.0) | NA |
| CTX-M-25 group | 0 (0.0) | 0 (0.0) | NA |
| CTX-M chimera | 0 (0.0) | 0 (0.0) | NA |
| GES | 0 (0.0) | 0 (0.0) | NA |

*K. quasipneumoniae*, *K. quasipneumoniae* subsp. *similipneumoniae*; NA, not applicable.

* P < 0.05.

**Table 4. Antimicrobial-resistant isolates of CTX-M-1 and CTX-M-9 group in ESBL-producing *K. pneumoniae*.**

| Antibiotic | Resistant isolates, no. (%) | | P value |
|---|---|---|---|
| | CTX-M-1 group ($n = 83$) | CTX-M-9 group ($n = 45$) | |
| β-Lactam combination | | | |
| Amoxicillin-clavulanate | 2 (2.4) | 2 (4.4) | 0.6 |
| Piperacillin-tazobactam | 16 (19.3) | 5 (11.1) | 0.3 |
| Cephem | | | |
| Ceftazidime | 61 (73.5) | 3 (6.7) | < 0.001* |
| Ceftriaxone | 83 (100) | 45 (100) | NA |
| Cefepime | 74 (89.2) | 20 (44.4) | < 0.001* |
| Cefmetazole | 0 (0.0) | 4 (8.9) | 0.01* |
| Carbapenem | | | |
| Imipenem | 0 (0.0) | 0 (0.0) | NA |
| Meropenem | 0 (0.0) | 0 (0.0) | NA |
| Aminoglycoside | | | |
| Gentamicin | 26 (31.3) | 21 (46.7) | 0.1 |
| Tobramycin | 23 (27.7) | 0 (0.0) | < 0.001* |
| Amikacin | 0 (0.0) | 0 (0.0) | NA |
| Fluoroquinolone | | | |
| Levofloxacin | 47 (56.6) | 15 (33.3) | 0.02* |
| Ciprofloxacin | 66 (79.5) | 18 (40.0) | < 0.001* |
| TMP-SMX | 74 (89.2) | 32 (71.1) | < 0.01* |

NA, not applicable; TMP-SMX, trimethoprim/sulfamethoxazole.

* $P < 0.05$.

rates. Conversely, cephems (excluding cefmetazole), fluoroquinolones, and TMP-SMX demonstrated high resistance rates. In both CTX-M-1 and CTX-M-9 groups, the resistance rates to amoxicillin-clavulanate piperacillin-tazobactam were <5% and <20%, respectively. The resistance rates to cefmetazole of the CTX-M-1 and CTX-M-9 groups were 0.0% and 8.9%, respectively. All isolates from both groups were susceptible to imipenem, meropenem, and amikacin.

## Characterization of AmpC-production in ESBL-producing *Klebsiella* and AmpC genotypes

The screening test results using cefoxitin disks showed that 16 *K. pneumoniae* (11.9% of *K. pneumoniae*) and one *K. quasipneumoniae* isolates were positive for AmpC screening (Table 1). AmpC-genotype test results showed that only one *K. pneumoniae* isolate was positive for the AmpC genotype (0.7% of *K. pneumoniae*, 95% CI 0–4.1), and this genotype was DHA. The DHA-positive isolate harbored the ESBL genotype consisting of coexisting CTX-M-9 and SHV. The remaining 15 AmpC screening positive *K. pneumoniae* were negative for all AmpC genotypes (11.9% of *K. pneumoniae*, 95% CI 6.4–17.7). One AmpC screening positive *K. quasipneumoniae* was negative for all AmpC genotypes.

## Discussion

We identified 135 *K. pneumoniae* and 4 *K. quasipneumoniae* ESBL-producing isolates from two regions in Japan. The predominant ESBL genotypes were the CTX-M group, with few instances of AmpC co-production.

*K. pneumoniae* is predominantly found among ESBL-producing *Klebsiella* isolates worldwide. In the United States [15] and Norway [8], the prevalence of ESBL-producing *K. pneumoniae* is between 98–99%, with ESBL-producing *K. quasipneumoniae* and *K. variicola* accounting for only 0.5–1.0%. In South and Southeast Asia, *K. pneumoniae* represents 92% of isolates, *K. quasipneumoniae* 7.6%, and *K. variicola* 0% [22]. Consistent with these reports, our study found *K. pneumoniae* to be the most predominant (97.1%), followed by *K. quasipneumoniae* (2.9%), with no *K. variicola* isolates identified. Both previous studies and our findings highlight that *K. pneumoniae* is a focal point in the antimicrobial resistance within the *Klebsiella* species. ESBL-producing *K. quasipneumoniae* is more prevalent in South and Southeast Asia than in the United States, Norway, and Japan, although the reasons for this disparity are unclear. Further research is needed to understand the distribution of ESBL-producing *K. quasipneumoniae*.

In our study, the CTX-M-1 group is the primary ESBL genotype in ESBL-producing *Klebsiella*. Our results are consistent with the predominance of the CTX-M-1 group in the United States [5], Canada [6], Europe [7,8], and Africa [9]. In terms of *Klebsiella* species, *K. pneumoniae* showed a significantly higher proportion of the CTX-M-1 group with SHV and TEM than *K. quasipneumoniae*, while *K. quasipneumoniae* showed a significantly higher proportion of the CTX-M-1 group only than *K. pneumoniae*. This result suggests differences in the ESBL genotypes prevalent in *K. pneumoniae* and *K. quasipneumoniae*. Furthermore, consistent with isolates collected in Japan in 2018 [10], our cohort showed a significant presence of the CTX-M-9 group. In contrast, the frequency of the CTX-M-9 group in ESBL-producing *Klebsiella* was 3–4% in the USA [5] and France [7] and 11–15% in Canada [6] and Africa [9]. Although these studies were carried out in different years, making simple comparisons impossible, ongoing surveillance of genotypes in ESBL-producing *Klebsiella* is essential to monitor the emergence of new resistant clones in different regions.

Carbapenems are effective against ESBL-producing isolates and play a vital role in treating the infections they cause [23–25]. In our cohort, ESBL-producing *Klebsiella* isolates demonstrated 100% sensitivity to carbapenems. Regarding cephalosporins, the CTX-M-1 groups exhibited higher resistance rates to ceftazidime and cefepime compared to the CTX-M-9 groups, with a particularly significant difference observed for ceftazidime. This discrepancy could potentially be linked to the prevalence of TEM. Most TEM enzymes are ceftazidimases, which possess strong capabilities in degrading ceftazidime [26]. The proportion of isolates harboring TEM was substantially higher in the CTX-M-1 group (56.3%) compared to the CTX-M-9 group (5.9%). Thus, disparities in the prevalence of TEM-carrying isolates may have contributed to the variations in resistance rates to ceftazidime between the CTX-M-1 and CTX-M-9 groups.

Amoxicillin-clavulanate, quinolones, and TMP-SMX are potential oral antimicrobial options [23–25]. The resistance rate to amoxicillin-clavulanate was 5.2% in ESBL-producing *K. pneumoniae* and 0% in ESBL-producing *K. quasipneumoniae*, indicating high susceptibility in both species. In contrast, resistance rates to levofloxacin, ciprofloxacin, and TMP-SMX were observed to be high. The CTX-M-1 group exhibited significantly higher resistance rates to these antimicrobials compared to the CTX-M-9 group. These findings indicate that quinolone and TMP-SMX resistance in ESBL-producing *Klebsiella* poses a significant challenge, especially within the CTX-M-1 group.

Only one isolate of ESBL- and AmpC-producing *K. pneumoniae* was identified, accounting for 0.7% of the isolates and carrying the DHA gene. This suggests that ESBL- and AmpC-producing *K. pneumoniae* are rare in Japan. In the United States [5] and Denmark [27], ESBL- and AmpC-producing *K. pneumoniae* prevalence was 3–4%. In contrast, a higher prevalence was observed in Iran [11], where 19 AmpC genes were detected in 52 ESBL-producing

*Klebsiella* isolates. These results indicate the regional differences in the epidemiology of ESBL- and Amp-producing *K. pneumoniae*. The presence of ESBL- and AmpC-producing *Klebsiella* raises concerns regarding limited antibiotic options for treating infections caused by these pathogens. Cefmetazole may be a carbapenem-sparing option for certain non-serious infections caused by ESBL-producing isolates, particularly urinary tract infections [23,28]. However, the presence of AmpC production confers resistance to cefmetazole in isolates. Given the rarity of ESBL- and AmpC-producing *K. pneumoniae* in Japan, cefmetazole may serve as a valuable alternative to carbapenems. Understanding the local distribution and characteristics of ESBL- and AmpC-producing *K. pneumoniae* is crucial for the development of effective antimicrobial therapy strategies.

Our study is subject to two primary limitations concerning the study site and the number of isolates. First, the study was conducted in two specific areas in Japan, and it's acknowledged that the proportion of *Klebsiella* species, antimicrobial resistance patterns, and ESBL genotypes can vary significantly across different countries and regions. Consequently, the trends observed in our study may not be generalizable to other settings. Second, there is a limitation due to the restricted number of *K. quasipneumoniae* isolates available for analysis. Isolates from the Sapporo Clinical Laboratory could not be analyzed due to lack of preservation prior to October 2023. This absence of data may have skewed the antimicrobial susceptibility patterns and ESBL genotypes, potentially impacting the overall distribution observed. Additionally, owing to the small sample size of *K. quasipneumoniae* and *K. variicola*, particularly regarding antimicrobial susceptibility and ESBL genotypes, we were unable to conduct a comprehensive comparison with *K. pneumoniae*. Given these limitations, future research incorporating additional areas and larger sample sizes, particularly for *K. quasipneumoniae* and *K. variicola*, is recommended.

Our study characterizes ESBL-producing *Klebsiella* isolates in Japan. Approximately 97% of these isolates are *K. pneumoniae*, with 62% harboring the CTX-M-1 group. AmpC carriage of ESBL *Klebsiella* is less than 1%, and carbapenems and cefmetazole are expected to be effective antibiotics against ESBL *Klebsiella* in Japan. However, *K. pneumoniae* with the CTX-M-1 group had a high rate of resistance to quinolones and TMP-SMX, and this multidrug-resistant pathogen is a concern in clinical practice. Given the limitations of the region and time period in this study, broader areas of research and ongoing monitoring are needed.

## Supporting information

**S1 Table. Data on the isolates of *Klebsiella* MALDI TypeR results.**
(XLSX)

**S2 Table. Data on the isolates of ESBL genotypes and antimicrobial susceptibilities.**
(XLSX)

## Acknowledgments

We thank the laboratory technicians at Kameda Medical Center and Sapporo Clinical Laboratory for their invaluable assistance in collecting the isolates.

## Author Contributions

**Conceptualization:** Naoki Watanabe.

**Data curation:** Naoki Watanabe.

**Formal analysis:** Naoki Watanabe, Miyuki Fujioka.

**Investigation:** Naoki Watanabe, Masahiko Ito.

**Methodology:** Naoki Watanabe, Tomohisa Watari, Yoshihito Otsuka, Masahiko Ito, Kazufumi Yamagata, Miyuki Fujioka.

**Project administration:** Yoshihito Otsuka.

**Resources:** Naoki Watanabe.

**Supervision:** Miyuki Fujioka.

**Validation:** Miyuki Fujioka.

**Visualization:** Naoki Watanabe.

**Writing – original draft:** Naoki Watanabe.

**Writing – review & editing:** Tomohisa Watari, Yoshihito Otsuka, Masahiko Ito, Kazufumi Yamagata, Miyuki Fujioka.

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
