## [Decision Letter · Decision Letter 0]

18 Mar 2024

PONE-D-24-04229Antimicrobial resistance and AmpC production in ESBL-producing Klebsiella pneumoniae and K. quasipneumoniae subsp. similipneumoniae: A retrospective study in Japanese clinical isolatesPLOS ONE

Dear Dr. Watanabe,

Thank you for submitting your manuscript to PLOS ONE. After careful consideration, we feel that it has merit but does not fully meet PLOS ONE’s publication criteria as it currently stands. Therefore, we invite you to submit a revised version of the manuscript that addresses the points raised during the review process.

Please submit your revised manuscript by May 02 2024 11:59PM. If you will need more time than this to complete your revisions, please reply to this message or contact the journal office at plosone@plos.org. Please include the following items when submitting your revised manuscript:A rebuttal letter that responds to each point raised by the academic editor and reviewer(s). You should upload this letter as a separate file labeled 'Response to Reviewers'.A marked-up copy of your manuscript that highlights changes made to the original version. You should upload this as a separate file labeled 'Revised Manuscript with Track Changes'.An unmarked version of your revised paper without tracked changes. You should upload this as a separate file labeled 'Manuscript'.

We look forward to receiving your revised manuscript.

Kind regards,

Mabel Kamweli Aworh, DVM, MPH, PhD. FCVSN

Academic Editor

PLOS ONE

Additional Editor Comments:

In addition to addressing the reviewer's comments, kindly highlight some recommendations based on the key findings of your study in the conclusion section.

Reviewers' comments:

Reviewer's Responses to Questions

**Comments to the Author**

1. Is the manuscript technically sound, and do the data support the conclusions?

Reviewer #1: Yes

Reviewer #2: Yes

Reviewer #3: Yes

Reviewer #4: Yes

2. Has the statistical analysis been performed appropriately and rigorously? 

Reviewer #1: Yes

Reviewer #2: Yes

Reviewer #3: No

Reviewer #4: Yes

3. Have the authors made all data underlying the findings in their manuscript fully available?

Reviewer #1: Yes

Reviewer #2: Yes

Reviewer #3: Yes

Reviewer #4: Yes

4. Is the manuscript presented in an intelligible fashion and written in standard English?

Reviewer #1: Yes

Reviewer #2: Yes

Reviewer #3: Yes

Reviewer #4: No

5. Review Comments to the Author

Reviewer #1: This is an important study with interesting findings, the manuscript is systematic and well put together, previous recommendations for corrections have been effected. In the discussion, it is always important to clearly state the health/public health implication of the work and how it has added to the body of knowledge. The minor issues raised in the review should be addressed. It is quite commendable and will add to the body of knowledge.

Reviewer #2: Great work.

Authors should consider that their work was done in a particular location and that should be stated where necessary, do not overgeneralize.

Further comments are included in the manuscript and a word document of same attached.

Reviewer #3: Lines 20-21

“The aim of this study is to evaluate the prevalence of various species, ESBL genotypes, 21 antimicrobial resistance patterns, and AmpC co-production in ESBL-producing Klebsiella”.

It is better for the authors to first introduce the basic concept of the work before going into the aim of the work. Also, the aim is too generic without a location attached to it.

Lins 54 to 56

“However, research on ESBL and AmpC co-producing K. pneumoniae is limited”. How is it limited? Internet search shows there are a lot of studies on this.

Lines 61-62

“the global epidemiology of ESBL-producing K. quasipneumoniae and K. variicola is ill62 understood” What do you mean by global epidemiology? This work is Japan focussed.

The work should be specific in its aims by attaching a location to it.

Lines 73-74

In this study, we 74 collected clinical isolates of ESBL-producing Klebsiella isolates and investigated.

Line 83-85

“In total, we collected 139 ESBL-producing Klebsiella isolates from two locations in Japan: Kameda Medical Center in the eastern part and Sapporo Clinical Laboratory in the northern part”

What is the basis for this? Which sampling technique was used?

Authors should present results in a way that shows significant differences or otherwise as set out under the statistics section.

The silence on significant differences that is obvious on the tables and body of results is obvious in the discussion too.

Reviewer #4: The authors provide useful information on the phenotypic and genotypic characterization of Klebsiella species from clinical isolates but needs to provide certain clarifications with major revisions. There are several repetitive presentations across the manuscript and authors should revisit their presentation style and revise the manuscript for a better outcome. Additionally, the manuscript will benefit from English editing for clarity.

- “The aim of this study is to evaluate the prevalence of various species, ESBL genotypes,

antimicrobial resistance patterns, and AmpC co-production in ESBL-producing Klebsiella” this is a bit confusing especially with the use of “prevalence” since the isolate were already identified and store as ESBL producing Klebsiella. I would suggest the use of “characterization” instead.

- The study provides interesting findings on the ESBL and AmpC genotypes in Klebsiella species from two healthcare facilities in Japan. The methodology session however needs to provide further information on the isolate sourcing. Were the isolates obtained from a repository and storage information? I am curious to know if there were some of the klebsiella species that were not confirmed even though they were stored as Klebsiella

- The authors need to provide acknowledgements for the facilities where the isolates were collected.

- The tables are followed by a verbatim report of the results presented in the Table. This is highly repetitive. The authors need to use either of the two methods.

- Table 3: it is not clear what the column “other” represents. If it is refereeing to the genotypes listed below it, then there is no need for this and it should be expunged. I suggest that the table should be presented in a clearer way. for example. To use “CTX-M-1group and CTX-M-1 alone” to differentiate where they occur tother with other genotypes and where it occurs alone rather than just “CTX-M-1” Also, “SHV group” should be reported in addition to “SHV alone”

- Table 5 does not add much value per se and should be expunged since the information has already been provided in line 258-264.

- The authors indicated that Phenotypic ESBL testing will be done in the methodology but made no mention of it in the result session or even in the supplementary table. It may be worthwhile to state how it compares with the ESBL genotyping.

- The discussions section needs to be improved to avoid repetition of results rather provide in-depth interrogation of the study findings for instance, the issue of co-production of ESBL and Amp C was briefly mentioned as a major complication of AMR but needs more elaborate discussion of the public health implication of co-production in clinical management of Klebsiella infections.

- The conclusion in its current state is a repetition of the results and lacks critical engagement with the study objectives, findings and implication.

Minor revisions

-The title gives prominence to K. quasipneumoniae subsp. Similipneumoniae even though there were very few isolates obtained and not much discussion around it in the discussion section

Pg 4 line 64- Correct to “Antimicrobial resistance pattern…”

- Pg 5 line 66: Add a full-stop after pneumonia

- Pg 5, line 72-80: Further information are needed in the methodology

-Pg 6 line 92: “further identification of Klebsiella species, detection……” This is confusing as it has been stated that MALDI-TOF was used for identification.

- Line 128: m-CIM…. Please write the full meaning of the acronym on first appearance

- Line 196-198: kindly expunge as this information has been provided in the methods section

-Line 213-215: same as above

Line 207: “The co-existing CTX….” This sentence lacks clarity. please rephrase.

-Line 219: “Within the CTX-M-9 group, coexisting CTX-M-9 and SHV were the most common genotype in K. pneumoniae (37/135, 27.4%).” This statement is unclear and needs to be rephrased for better clarity.

- Line 244-251- This paragraph does not seem to add any real value but merely states what can be clearly observed from the table> the authors need to choose either tables or description

6. PLOS authors have the option to publish the peer review history of their article (what does this mean?). If published, this will include your full peer review and any attached files.

Reviewer #1: **Yes: **Rahab Charles-Amaza

Reviewer #2: No

Reviewer #3: No

Reviewer #4: **Yes: **Emelda Chukwu

---

## [Author Response · Author response to Decision Letter 0]

5 Apr 2024

We thank the reviewers for their valuable comments. Their suggestions have been incorporated into our revised manuscript. We presented our responses to each comment and the corresponding revisions to the manuscript in the "Response to Reviewers" document.

---

## [Decision Letter · Decision Letter 1]

24 Apr 2024

Antimicrobial resistance and AmpC production in ESBL-producing Klebsiella pneumoniae and Klebsiella quasipneumoniae: A retrospective study in Japanese clinical isolates

PONE-D-24-04229R1

Dear Dr. Watanabe,

We’re pleased to inform you that your manuscript has been judged scientifically suitable for publication and will be formally accepted for publication once it meets all outstanding technical requirements.

Kind regards,

Mabel Kamweli Aworh, DVM, MPH, PhD. FCVSN

Academic Editor

PLOS ONE

Additional Editor Comments (optional):

Reviewers' comments:

Reviewer's Responses to Questions

**Comments to the Author**

1. If the authors have adequately addressed your comments raised in a previous round of review and you feel that this manuscript is now acceptable for publication, you may indicate that here to bypass the “Comments to the Author” section, enter your conflict of interest statement in the “Confidential to Editor” section, and submit your "Accept" recommendation.

Reviewer #2: All comments have been addressed

Reviewer #4: All comments have been addressed

2. Is the manuscript technically sound, and do the data support the conclusions?

Reviewer #2: Yes

Reviewer #4: Yes

3. Has the statistical analysis been performed appropriately and rigorously? 

Reviewer #2: Yes

Reviewer #4: Yes

4. Have the authors made all data underlying the findings in their manuscript fully available?

Reviewer #2: Yes

Reviewer #4: Yes

5. Is the manuscript presented in an intelligible fashion and written in standard English?

Reviewer #2: Yes

Reviewer #4: Yes

6. Review Comments to the Author

Reviewer #2: The authors have done a good job by attending to the concerns raised earlier. The findings are essential in advancing knowledge.

Reviewer #4: (No Response)

7. PLOS authors have the option to publish the peer review history of their article (what does this mean?). If published, this will include your full peer review and any attached files.

Reviewer #2: No

Reviewer #4: No

---

## [Editor Report · Acceptance letter]

2 May 2024

PONE-D-24-04229R1 

PLOS ONE

Dear Dr. Watanabe, 

I'm pleased to inform you that your manuscript has been deemed suitable for publication in PLOS ONE. Congratulations! Your manuscript is now being handed over to our production team.

Kind regards, 

on behalf of

Dr. Mabel Kamweli Aworh 

Academic Editor

PLOS ONE